# Ectodomain shedding of EGFR ligands serves as an activation readout for TRP channels

**Manae Tatsumi**[1], **Takayuki Kishi**[1], **Satoru Ishida**[1], **Hiroki Kawana** [2], **Akiharu Uwamizu**[2], **Yuki Ono**[1], **Kouki Kawakami**[1], **Junken Aoki**[2], **Asuka Inoue** [1]*

1 Molecular and Cellular Biochemistry, Graduate School of Pharmaceutical Sciences, Tohoku University, Sendai, Japan, 2 Department of Health Chemistry, Graduate School of Pharmaceutical Sciences, The University of Tokyo, Bunkyo-ku, Tokyo, Japan

* iaska@tohoku.ac.jp

## Abstract

Transient receptor potential (TRP) channels are activated by various extracellular and intracellular stimuli and are involved in many physiological events. Because compounds that act on TRP channels are potential candidates for therapeutic agents, a simple method for evaluating TRP channel activation is needed. In this study, we demonstrated that a transforming growth factor alpha (TGFα) shedding assay, previously developed for detecting G-protein–coupled receptor (GPCR) activation, can also detect TRP channel activation. This assay is a low-cost, easily accessible method that requires only an absorbance microplate reader. Mechanistically, TRP-channel-triggered TGFα shedding is achieved by both of a disintegrin and metalloproteinase domain-containing protein 10 (ADAM10) and 17 (ADAM17), whereas the GPCR-induced TGFα shedding response depends solely on ADAM17. This difference may be the result of qualitative or quantitative differences in intracellular $Ca^{2+}$ kinetics between TRP channels and GPCRs. Use of epidermal growth factor (EGF) and betacellulin (BTC), substrates of ADAM10, improved the specificity of the shedding assay by reducing background responses mediated by endogenously expressed GPCRs. This assay for TRP channel measurement will not only facilitate the high-throughput screening of TRP channel ligands but also contribute to understanding the roles played by TRP channels as regulators of membrane protein ectodomain shedding.

## Introduction

Transient receptor potential (TRP) channels comprise the largest cation-channel family and are regulated by various types of stimuli. In humans, 28 TRP channel subtypes have been identified, divided into seven subfamilies: TRPA, TRPC, TRPM, TRPML, TRPN, TRPP, and TRPV [1]. TRPV1, TRPA1, and TRPM8 are canonical members that were first identified as nociceptive receptors for capsaicin, allyl isothiocyanate (AITC), and menthol, respectively [2–5], compounds that act on nerve cells expressing TRP channels found in the oral mucosa, resulting in the respective perceptions of heat, pungent irritation, and coolness [6, 7]. In addition to nerve cells, TRP channels are expressed in many cell types, such as keratinocytes and immune cells,

**Data Availability Statement:** All data are provided in the paper and its Supporting Information file.

**Funding:** A.I. was supported by Japan Society for the Promotion of Science (JSPS) KAKENHI grants

16H01377, 21H04791, 21H05113, JPJSBP120213501 and JPJSBP120218801; FOREST Program JPMJFR215T and JST Moonshot Research and Development Program JPMJMS2023 from Japan Science and Technology Agency (JST); Takeda Science Foundation; Uehara Memorial Foundation. M.T. received JSPS KAKENHI 22J10475. The funders had no role in study design, data collection and analysis, decision to publish, or preparation of the manuscript.

**Competing interests:** The authors have declared that no competing interests exist.

and are involved in various functions, including hair follicle formation and inflammation [8–11], making them attractive targets for drug discovery [12, 13].

Compounds acting on TRP channels can be evaluated by detecting the activation of TRP channel-induced intracellular signaling pathways. In most cases, TRP channel activation is assessed by measuring increases in intracellular $Ca^{2+}$ concentrations using calcium mobilization assays or patch-clamp techniques; however, these experiments require expensive, specialized equipment (e.g., fluorescence measurement devices with liquid handling and patch-clamp amplifiers, respectively). Therefore, the development of a low-cost and easy-to-access method for evaluating TRP channel activation will be useful for researchers, especially those who have limited access to specialized equipment, and may broaden the TRP channel field.

We hypothesized that a transforming growth factor alpha (TGFα) shedding assay, which we have previously developed for detecting G-protein-coupled receptor (GPCR) activation [14], has the potential to detect TRP channel activation. The TGFα-shedding assay is based on the ectodomain shedding of the membrane-bound proform of alkaline phosphatase–tagged TGFα (pro-AP-TGFα). The activation of Gq-coupled or G12-coupled GPCRs induces TGFα ectodomain shedding via a disintegrin and metalloproteinase domain-containing protein 17 (ADAM17), which cleaves pro-TGFα and other membrane proteins. ADAM17 is activated by protein kinase C (PKC) and intracellular $Ca^{2+}$ elevation [14–17]. Because TRP channels also trigger intracellular $Ca^{2+}$ influx, we examined whether TRP channel activation can be detected using the TGFα-shedding assay.

In addition to ADAM17, ADAM10 also cleaves pro-TGFα, but the upstream mechanism associated with ADAM10 activation reportedly differs from that associated with ADAM17 activation. ADAM10 induces a TGFα-shedding response upon stimulation with calcium ionophores but is insensitive to the PKC activator tetradecanoyl phorbol acetate (TPA) that stimulates ADAM17 [17]. Our previous study showed that ADAM17 knockdown greatly suppressed the GPCR-induced TGFα-shedding response, suggesting that this response is independent of ADAM10 [14]. Activation of ADAM10 requires an increase in $Ca^{2+}$ concentration, such as that induced by calcium ionophores, but the upstream physiological factors that lead to ADAM10 activation are poorly understood. Therefore, we examined whether TRP channels are upstream factors that activate ADAM10.

Here we demonstrate the ability of the TGFα-shedding assay to detect the activation of canonical, representative members of the TRP channel family. Interestingly, TRP channels induce not only the ADAM17-dependent but also the ADAM10-dependent TGFα-shedding response. We took advantage of this difference to minimize nonspecific responses mediated by endogenously expressed GPCRs by using epidermal growth factor (EGF) and betacellulin (BTC), substrates of ADAM10, as reporters in place of TGFα.

## Materials and methods

### Reagents and plasmids

Chemicals and reagents were purchased from Wako Pure Chemical Industries unless otherwise noted. Stealth small interfering RNA (siRNA) duplexes against mRNA encoding ADAM10 (target sequences are listed in Supplementary Information) and Stealth negative control siRNAs were purchased from Invitrogen. All TRP channels and GPCRs used in this study were of human origin and did not contain epitope tags. The sequences for TRP channels and GPCRs were cloned into mammalian expression vectors pcDNA3.1 (Invitrogen) and pCAGGS (gift from J. Miyazaki, Osaka University), respectively. A plasmid encoding alkaline phosphatase (AP)-TGFα was described previously [18]. Plasmids encoding AP-epidermal

growth factor (EGF) and AP-betacellulin (BTC) were gifts from Shigeki Higashiyama, Ehime University [19].

## Cell culture and transfection

Parent and ADAM17-deficient HEK293 cells were maintained in Dulbecco's modified Eagle medium (DMEM, Nissui Pharmaceutical) supplemented with 10% fetal bovine serum (Gibco), 100 U/ml penicillin (Sigma-Aldrich), and 100 µg/ml streptomycin (Gibco) (complete DMEM) in a 37˚C incubator with 5% $CO_2$. Transfection of plasmid DNAs was performed by lipofection reagent, polyethylenimine solution (PEI Max, Polysciences). Typically, cells were seeded in each well of a 12-well culture plate at a cell density ranging from $2 \times 10^5$ to $3 \times 10^5$ cells/mL in 1 mL complete DMEM and cultured for 1 day in a 37˚C incubator with 5% $CO_2$. For transfection, plasmid solution (see each assay condition below) was diluted in 50 µL Opti-MEM (Gibco) and mixed with 2.5 µL of 1 mg/mL PEI solution in 50 µL Opti-MEM. Cells were incubated for 1 day after transfection before performing any assays. Transfection of siRNA was performed using Lipofectamine RNAiMAX (Invitrogen). Cells were seeded in each well of a 12-well culture plate at a cell density of $1 \times 10^5$ cells/mL in 1 mL complete DMEM and cultured for 1 day in a 37˚C incubator with 5% $CO_2$. For transfection, 1.2 µL of 10 µM siRNA was diluted in 100 µL Opti-MEM and mixed with 2 µL RNAiMAX in 100 µL Opti-MEM. Cells were incubated for 1 day before performing plasmid transfection. Immediately prior to plasmid transfection, the cell supernatant was removed by aspiration, and fresh complete DMEM was added. Transfection then proceeded as described above.

## TGFα-shedding assay

The TGFα-shedding assay was performed as described previously, with minor modifications [14]. Plasmid transfection was performed in a 12-well plate using a mixture of 250 ng plasmid encoding AP-TGFα (or AP-EGF or AP-BTC) and 100 ng plasmid encoding the receptor. After 1 day of culture, transfected cells were harvested by trypsinization, pelleted by centrifugation at $190 \times g$ for 5 min, and suspended in 3.5 mL Hank's Balanced Salt Solution (HBSS) containing 5 mM HEPES (pH 7.4). After incubation for 15 min at room temperature, cells were centrifuged at $190 \times g$ for 5 min, and cell pellets were suspended in 3.5 mL HBSS. The resuspended cells were plated in a 96-well plate at 90 µl per well (typically 24 total wells [$8 \times 3$]) and placed in a 37˚C incubator with 5% $CO_2$ for 30 min. After incubation, 10 µl of 10× compounds were added to each well and incubated for 1 h at 37˚C in 5% $CO_2$. Plates were centrifuged at $190 \times g$ for 2 min. After centrifugation, 80 µl of supernatant from each well was transferred to a clean well in a new 96-well plate, leaving attached cells and 20 µl supernatant in the original well. An 80 µl volume of para-nitrophenyl phosphate (p-NPP) solution (10 mM p-NPP; 40 mM Tris-HCl, pH 9.5; 40 mM NaCl; 10 mM $MgCl_2$) was then added to each well of both the supernatant plate and the cell plate. Absorbance at 405 nm ($Abs_{405}$) was obtained for both plates before (background) and after 1-h incubation at 37˚C using a microplate reader (SpectraMax 340 PC384, Molecular Devices). TGFα release was calculated as described in the Results. To evaluate TRPV1 antagonism, cells (plated in 80 µL per well) were pretreated with various concentrations of compounds 10 min before stimulation with capsaicin (100 nM). To evaluate TRPV1 inverse agonism, cells were incubated with capsazepine in the absence of capsaicin.

## Generation of ADAM17-deficient cells

CRISPR-based targeted gene depletion was performed as described previously, with minor modifications [20]. In detail, ADAM17-deficient HEK293 cells were generated by the

CRISPR/Cas9 system to mutate the gene encoding ADAM17 in parent HEK293 cells. An sgRNA construct targeting the gene encoding ADAM17 was designed using a CRISPR design tool (http://crispr.mit.edu) such that an SpCas9-mediated DNA cleavage site (3-bp upstream of the protospacer adjacent motif [PAM] sequence [NGG]) encompassed a restriction enzyme recognition site. The designed sgRNA sequence for ADAM17, including the SpCas9 PAM sequence, was 5′-GACCATTGAAAGTAAGGCCC-3′ (the Hae III restriction enzyme site is underlined). The designed sgRNA sequence was inserted into the Bbs I site of the pSpCas9 (BB)-2A-GFP (PX458) vector (a gift from Feng Zhang at the Broad Institute; Addgene plasmid No. 48138). The correct insertion of the sgRNA sequence was verified by Sanger sequencing (Fasmac). To generate ADAM17-deficient cells, HEK293 cells were seeded into a 10-cm culture dish and incubated for 24 h before transfection. The PX458 plasmid encoding the sgRNA and SpCas9-2A-GFP was transfected into cells using Lipofectamine 2000 (Thermo Fisher Scientific). After 3 days, the cells were harvested and processed for the isolation of GFP-positive cells (~5% of cells) using a fluorescence-activated cell sorter (SH800; Sony). After the expansion of clonal cell colonies using a limiting dilution method, clones were analyzed for the incorporation of the mutation in the targeted gene by restriction enzyme digestion. Candidate clones that harbored restriction enzyme–resistant PCR fragments were further assessed for genomic DNA alterations by TA cloning [21]. The PCR primers used to amplify the sgRNA-targeted sites were as follows: 5′-CCATAACTCCAGGGTGGCTC-3′ and 5′-GAGAGACTCCT CACCTGCAC-3′.

## Data analysis

Concentration–response curves were fitted for all data using the Nonlinear Regression: Variable slope (four parameters) function in the GraphPad Prism 9 software (GraphPad), with the setting of absolute Hill Slope values less than 2. Sigmoid maximum effect ($E_{max}$), the negative log of the half-maximal excitation concentration ($pEC_{50}$, a parameter for agonism), and the negative log of the half-maximal inhibitory concentration ($pIC_{50}$, a parameter for antagonism and inverse agonism) were obtained. The details regarding normalization and replicates for each experiment are described in the figure legends.

## Results

### TRP channel activation is indicated by TGFα ectodomain shedding

First, we tested whether TRP channel activation triggers TGFα-shedding responses. We transfected plasmids encoding the AP-TGFα reporter with or without TRPV1-encoding plasmids in HEK293 cells (Fig 1A and 1B). After 24 h, we reseeded the transfected cells and stimulated them for 1 h with capsaicin, a prototypical TRPV1 agonist. If TRPV1 activation by capsaicin induces ectodomain shedding of AP-TGFα, we would expect to detect AP in the supernatant (Fig 1A and 1B). We isolated the supernatant and added $p$-NPP, a substrate for AP, to both the supernatant and the cells. AP activity can be evaluated by measuring para-nitrophenol ($p$-NP) production at $Abs_{405}$ using an absorbance microplate reader (Fig 1B). Released AP-TGFα (%) can be determined by calculating the ratio of AP activity in the supernatant to the total AP activity (Fig 1B). We observed a TGFα-shedding response in a capsaicin concentration–dependent manner in TRPV1-transfected cells but not in mock-transfected cells (Fig 1C). The $pEC_{50}$ value calculated from this concentration–response curve (Fig 1C) was 7.99 ± 0.05. We then expressed three other TRP channels, namely TRPA1, TRPM8, and TRPV3, and measured activation upon stimulation by their respective ligands, AITC, menthol, and 2-aminoethoxydiphenyl borate (2-APB). TGFα-shedding responses were observed under all tested receptor-transfected conditions (Fig 1D–1F). At high concentrations of AITC and 2-APB, the TGFα-

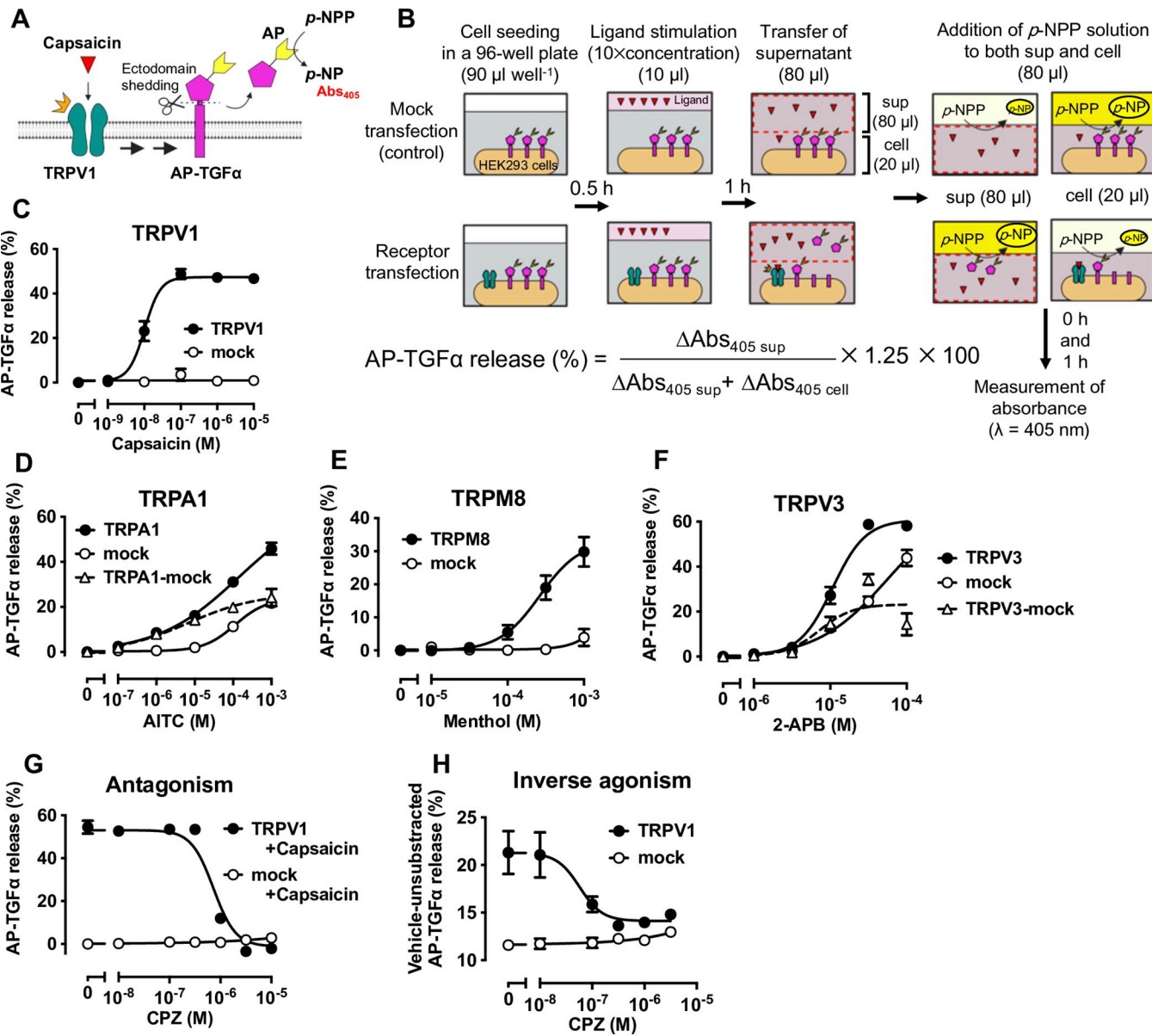

**Fig 1. TGFα-shedding assay can evaluate TRP channel activation.** (A) Schematic representation of the TGFα-shedding assay for detecting transient receptor potential (TRP) channel activation. TRPV1 is activated upon capsaicin stimulation, inducing AP-TGFα ectodomain shedding. Released AP-TGFα can be quantified by measuring AP activity in the supernatant based on the production of para-nitrophenol ($p$-NP) from para-nitrophenyl phosphate ($p$-NPP). (B) Schematic of the assay protocol: HEK293 cells transiently expressing AP-TGFα with or without TRP channel expression are reseeded onto 96-well plates and stimulated with a ligand. After the supernatant (sup) is transferred to a blank plate, AP-TGFα release is quantified by a colorimetric reaction to measure AP activity, using $p$-NPP as a substrate. AP-TGFα release (%) is calculated as the ratio of AP activity in the supernatant to the total AP activity. $\Delta Abs_{405}$ was calculated by subtracting the absorbance at 405 nm measured at 0 h [$Abs_{405\ (0\ h)}$] from the absorbance at 405 nm measured at 1 h [$Abs_{405\ (1\ h)}$], using 1.25 as a correction factor for the amount of supernatant transferred (80 of 100 μL). See Methods for details. (C) Concentration–response curve for the TGFα-shedding responses induced by TRPV1 activation upon capsaicin stimulation. The vehicle-treated condition is set as the baseline. Mock-transfected cells expressing only the AP-TGFα reporter were used as a control. (D–F) Concentration–response curves for the TGFα-shedding responses induced by TRPA1 (C), TRPM8 (D), and TRPV3 (E) activation upon AITC, menthol, and 2-APB stimulation, respectively. (G, H) Evaluation of antagonist activity (G) and inverse agonist activity (H) for capsazepine (CPZ). CPZ antagonism was examined in the presence of 100 nM capsaicin. Note that inverse agonism is shown without subtracting the vehicle-treated basal responses. In all panels, the symbols and error bars represent the mean and SEM, respectively, for three independent experiments performed in triplicate. For many data points, the vertical error bars are smaller than the symbols and, thus, are not visible.

shedding response was observed in mock-transfected cells, suggesting that these chemicals have off-target activities in HEK293 cells. However, the TGFα-shedding responses in TRPA1- and TRPV3-expressing cells were higher than those in mock-transfected cells. We evaluated the TRP channel–dependence of signals by subtracting the response in mock-transfected cells from the response in TRP channel–expressing cells (Fig 1D and 1F). The pEC$_{50}$ values in TRPA1-, TRPM8-, and TRPV3-expressing cells were 5.17 ± 0.34, 3.58 ± 0.13, and 5.14 ± 0.22, respectively. The pEC50 values for all four types of channels were equivalent to or one order of magnitude greater than the values reported using other established methods [22–26]. These results demonstrate that the TGFα-shedding assay sensitively detects agonist-induced activation of TRP channels.

Next, we attempted to assess whether antagonist activity is measurable. We measured the antagonist activity of capsazepine, a TRPV1 antagonist, in the presence of 100 nM of capsaicin [27]. The capsaicin-induced TGFα-shedding response was inhibited by increasing concentrations of capsazepine in TRPV1-transfected cells (Fig 1G). In mock-transfected cells, the shedding response was not observed at any tested concentration. The pIC$_{50}$ value calculated from this concentration–response curve (Fig 1G) was 6.14 ± 0.064, which is consistent with a previous report [28], indicating that this assay can be used to evaluate antagonist activity.

Finally, we evaluated the inverse agonist activity of capsazepine, which refers to the inhibitory effect of capsazepine against spontaneous TRPV1 activation. Many TRP channels exhibit spontaneous activity that is observed just by expressing them in cultured cells, which is caused by constitutive activation of upstream factors of TRP channels [29]. This spontaneous activity causes chronic pain and degeneration in nerves and other cells, making it an important therapeutic target [30, 31]. We incubated TRPV1-expressing or mock-transfected cells with increasing concentrations of capsazepine for 1 h and measured the TGFα-shedding response. TRPV1-expressing cells showed a TGFα-shedding response without ligand stimulation, and the addition of capsazepine suppressed this response to a level similar to the level observed for mock-transfected cells (Fig 1H, the TGFα-shedding response is shown without subtracting the basal response). This result indicates that the TGFα-shedding assay can be used to evaluate the spontaneous activation of TRP channels and the inverse agonist activity of ligands. Therefore, the TGFα-shedding assay is useful for evaluating TRP channel activity.

## TGFα-shedding response induced by TRPV1 activation depends on ADAM10 and ADAM17

We next investigated whether TRP channels and GPCRs induce the TGFα-shedding response through a shared sheddase. In a previous study, the GPCR-induced TGFα-shedding response was greatly suppressed by the siRNA-mediated knockdown of ADAM17 [14]. In the present study, the elimination of ADAM17 from HEK293 cells (ΔADAM17 cells; S1 Fig) completely abolished the TGFα-shedding response induced by the activation of the Gq-coupled histamine H1 receptor (H1R; Fig 2A), indicating that the GPCR-induced TGFα-shedding response is totally dependent on ADAM17. Stimulation of PKC by TPA induced a robust TGFα-shedding response, which was silenced in ΔADAM17 cells, suggesting that GPCR-induced TGFα shedding solely depends on the Gq–PKC–ADAM17 axis (Fig 2B).

We then examined whether TRPV1-induced TGFα shedding is mediated by ADAM17. Unlike H1R or TPA stimulation, capsaicin-induced TRPV1 stimulation remained capable of inducing a TGFα-shedding response in the ΔADAM17 cells, although the potency of capsaicin was lower in ΔADAM17 cells than in parent HEK293 cells (Fig 2C). This result indicates that other sheddases are involved in the TRPV1-induced TGFα-shedding response. Potent TGFα cleavage activity is also characteristic of ADAM10 [16, 32]. The TRPV1-induced TGFα-

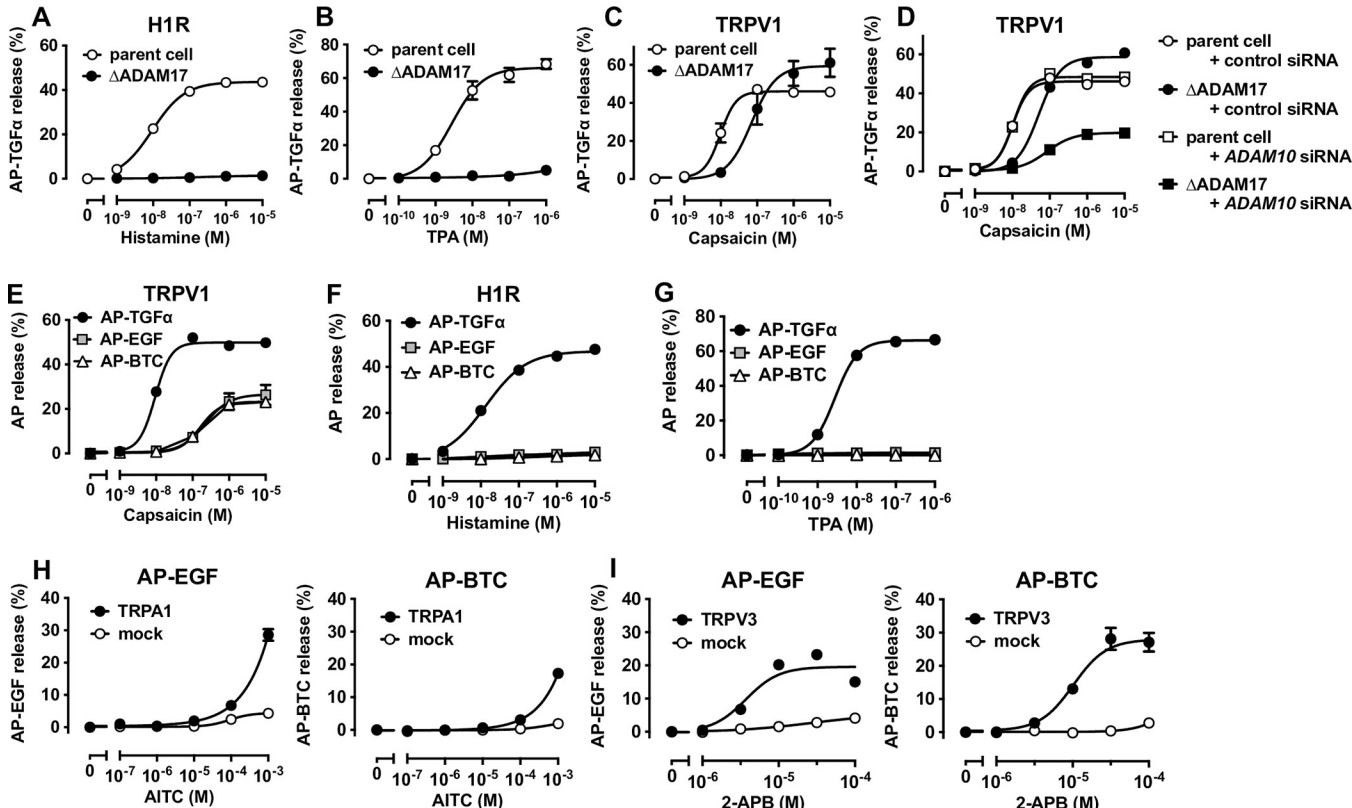

**Fig 2. TRP channel activation induces ectodomain shedding of EGFR ligands via both ADAM10 and ADAM17.** (A–C) Concentration–response curves showing the TGFα-shedding response in ADAM17-deficient HEK293 cells (ΔADAM17 cells). H1R (A) and TRPV1 (C) were evaluated as representatives of Gq-coupled GPCRs and TRP channels, respectively. TPA-induced shedding responses (B) were evaluated in cells expressing only AP-TGFα without receptors. (D) Concentration–response curve of TGFα-shedding responses induced by TRPV1 activation in ADAM10-siRNA-transfected parent cells or ΔADAM17 cells. (E–G) Comparisons of shedding responses induced by TRPV1 (E) and H1R (F) activation using AP-EGF and AP-BTC as reporters. The TPA-induced shedding response (G) was evaluated in cells expressing only AP-TGFα, AP-EGF, or AP-BTC, without receptors. (H, I) Concentration–response curves for the EGF- and BTC-shedding responses induced by TRPA1 (H) and TRPV3 (I) activation. In all figures, the symbols and error bars represent the mean and SEM, respectively, for three independent experiments performed in triplicate. For many data points, the vertical error bars are smaller than the symbols and, thus, are not visible.

shedding response was evaluated in parent and ΔADAM17 cells transfected with ADAM10-siRNA or non-target siRNA. The TGFα-shedding response was not suppressed in ADAM10-siRNA–transfected parent cells but was greatly suppressed in ADAM10-siRNA–transfected ΔADAM17 cells (Fig 2D). These results indicate that TGFα-shedding response induced by TRP channels involves both the ADAM17 and ADAM10 (and possibly other ADAMs) pathways, whereas the GPCR-induced response depends solely on ADAM17.

Membrane protein substrates with high ADAM10 selectivity may be useful for measuring TRP channel ligands that induce non-specific responses, which may be mediated by endogenously expressed GPCRs, PKCs, or their signaling axes. EGF and BTC are known ADAM10 substrates [19, 33]. In parent cells, we expressed AP-tagged EGF (AP-EGF) or BTC (AP-BTC) in place of AP-TGFα and evaluated their shedding responses. TRPV1 activation induced both shedding responses, whereas H1R and PKC activation induced neither (Fig 2E–2G). Although the pEC$_{50}$ and E$_{max}$ values for AP-BTC and AP-EGF were both lower than those for AP-TGFα, AP-BTC and AP-EGF are useful for selectively detecting TRP channel-induced responses. We next examined whether the AP-EGF and the AP-BTC reporters lower the background signals observed for the TRPA1 and TRPV3 ligands. As shown in Fig 1C and 1E,

AITC and 2-APB induced a TGFα-shedding response in mock-transfected cells. In TRPA1- and TRPV3-expressing cells, AITC and 2-APB, respectively, induced AP-EGF- and AP-BTC-shedding responses in concentration-dependent manners, whereas no shedding responses were observed in mock-transfected cells (Fig 2H and 2I). These results indicate that AP-EGF and AP-BTC are useful for TRP-selective measurements, with potentially low background responses. Therefore, unlike GPCRs, TRP channels can activate ADAM10, and this different ectodomain shedding mechanism allows for shedding assay improvements that facilitate the selective detection of TRP channel activation.

## Discussion

In this study, we showed that the TGFα-shedding assay detects TRP channel activation and can be used to evaluate the agonist, antagonist, and inverse agonist activities of compounds. This assay has several advantages. First, it detects amplified signals based on the accumulation of pro-AP-TGFα in the supernatant and, thus, can detect the basal activity of TRP channels. The conventional $Ca^{2+}$ mobilization assay is unsuitable for evaluating inverse agonistic activity because it captures a transient intracellular event. Second, the TGFα-shedding assay is less costly than traditional $Ca^{2+}$ mobilization assay and patch-clamp method and can be performed with an easy-to-access absorbance microplate reader, a simple transient expression system, and low-cost reagents (*p*-NPP). One disadvantage of the TGFα-shedding assay includes the requirement for high exogenous AP-TGFα reporter expression, making this assay unsuitable for the analysis of primary cultured cells, in which the induction of exogenous expression can be challenging. We also recommend using AP-EGF and AP-BTC in place of AP-TGFα as reporters for measurements of TRP channel-ligand pairs to eliminate nonspecific responses.

Interestingly, ADAM10 is uniquely activated by TRP channels, despite both TRP channels and GPCRs inducing intracellular $Ca^{2+}$ influx (Fig 2A–2D). This was evidenced by ectodomain shedding of EGF and BTC, substrates of ADAM10, which was triggered only by the downstream signal of TRP channels (Fig 2E–2I). The increase in intracellular $Ca^{2+}$ concentration induced by TRP channel activation is high and long-lasting, whereas that induced by GPCRs is oscillatory and transient [34–36]. Previous reports showed that $Ca^{2+}$-induced ADAM10 activation is totally dependent on anoctamin 6 (ANO6), a $Ca^{2+}$-sensitive phosphatidylserine scramblase [37, 38]. Based on these studies, we speculate that TRP channels activate ADAM10 via ANO6. The lack of GPCR-induced ADAM10 activation (Fig 2A) indicates no involvement of ANO6. ANO6 activation requires a high intracellular $Ca^{2+}$ concentration ($[Ca^{2+}]i > 1$ μM) [39–41], and the increase in $Ca^{2+}$ concentration or the duration of increased $Ca^{2+}$ induced by GPCR activation is likely insufficient to activate ANO6. Although ADAM17 is activated by ANO6 [37], GPCR-triggered ADAM17 activation depends solely on PKC [14]. Thus, the distinct kinetics of intracellular $Ca^{2+}$ concentrations associated with TRP channel and GPCR activation likely underlie their different signaling outcomes (Fig 3).

ADAM10 and ADAM17 activation induced by the TRP channels may occur under both physiological and pathological conditions. A previous report showed that TRPV3 induces a TGFα-shedding response via activation of ADAM17, promoting hair follicle formation [9]; however, the physiological relationship between ADAM10 and TRP channels remains unknown. ADAM10 is associated with Alzheimer's disease (AD) and is a potential therapeutic target [42–44]. ADAM10 has α-secretase activity, which prevents amyloid-β formation [45, 46], and loss-of-function mutations in ADAM10 have been reported in AD patients, indicating that ADAM10 activation may be useful for the treatment of AD [47]. TRP channels, as well as ADAM10, are expressed in central nervous system neurons and are associated with psychiatric disorders [48, 49]. We believe that testing whether ADAM10 is activated by TRP channels

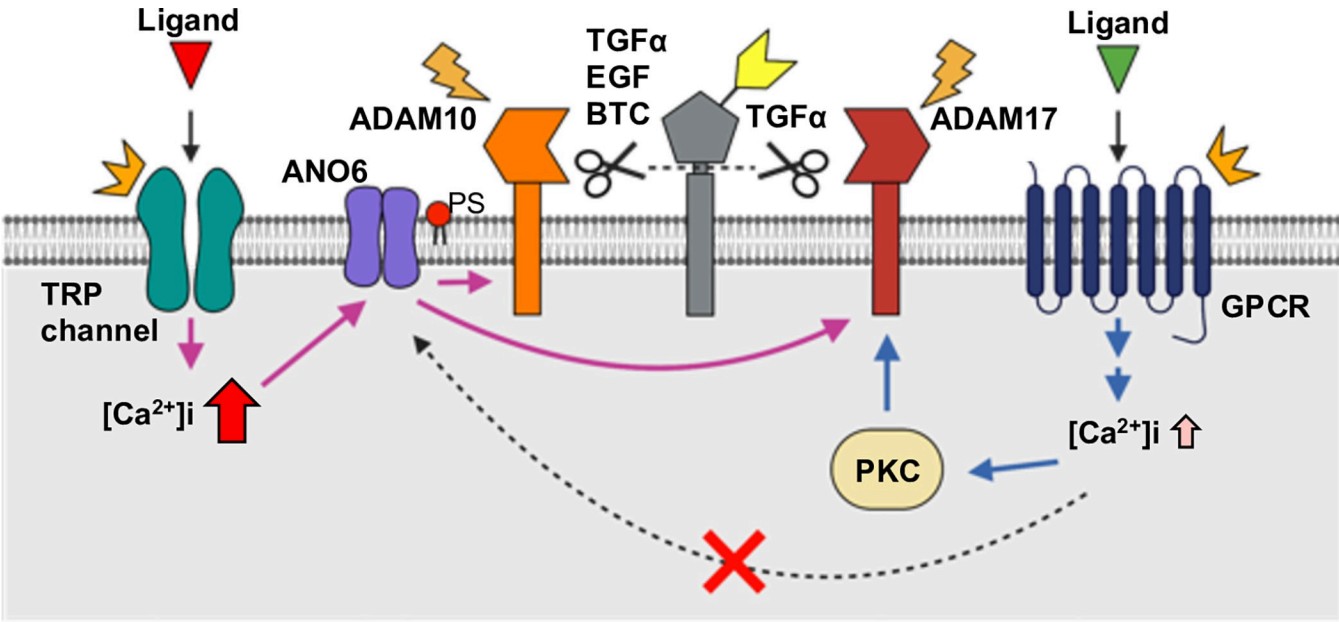

**Fig 3. Possible mechanisms underlying ectodomain shedding of EGFR ligands by TRP channel and GPCR activation.** Putative molecular mechanisms for ectodomain shedding of EGFR ligands by TRP channels and Gq-coupled GPCRs activation. ADAM, a disintegrin and metalloproteinase domain-containing protein; ANO6, anoctamin 6; BTC, betacellulin; EGF, epidermal growth factor; GPCR, G protein–coupled receptor; PKC, protein kinase C; PS, phosphatidylserine; TGFα, transforming growth factor alpha; TRP, transient receptor potential.

under physiological conditions is worthwhile, in addition to determining whether targeting TRP channels to activate ADAM10 is potentially useful for the treatment of AD.

## Supporting information

**S1 Fig. Genomic sequences of ADAM17-deficient HEK293 cell line.** The sgRNA-target sequence is underlined. The arrow indicates a putative double-stranded break site. The restriction enzyme site (Hae III) is highlighted in red.
(TIF)

**S1 Data. The numerical values underlying Figs 1 and 2.**
(XLSX)

## Acknowledgments

We thank Shigeki Higashiyama (Ehime University) for the AP-fused constructs; Makoto Arita (Keio University) for the helpful information on the TRP channel and TGFα shedding response; Makoto Tominaga (National Institute for Physiological Sciences) for the helpful discussion on TRP channel assays; Kayo Sato, Shigeko Nakano and Ayumi Inoue (Tohoku University) for their assistance in plasmid preparation; Tatsuya Ikuta (Tohoku University) for helpful discussion and manuscript editing.

## Author Contributions

**Conceptualization:** Asuka Inoue.

**Funding acquisition:** Junken Aoki, Asuka Inoue.

**Investigation:** Manae Tatsumi, Takayuki Kishi, Satoru Ishida, Hiroki Kawana, Akiharu Uwamizu, Yuki Ono, Kouki Kawakami, Asuka Inoue.

**Methodology:** Asuka Inoue.

**Supervision:** Junken Aoki, Asuka Inoue.

**Writing – original draft:** Manae Tatsumi, Asuka Inoue.

**Writing – review & editing:** Manae Tatsumi, Asuka Inoue.

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
