## [Decision Letter · Decision Letter 0]

6 Dec 2022

PONE-D-22-29508Ectodomain shedding of EGFR ligands serves as an activation readout for TRP channelsPLOS ONE

Dear Dr. Inoue,

Thank you for submitting your manuscript to PLOS ONE. After careful consideration, we feel that it has merit but does not fully meet PLOS ONE’s publication criteria as it currently stands. Therefore, we invite you to submit a revised version of the manuscript that addresses the points raised during the review process.

The issues pointed by one of reviewers should be discussed in the corresponding section of the MS

We look forward to receiving your revised manuscript.

Kind regards,

Agustín Guerrero-Hernandez

Academic Editor

PLOS ONE

Journal Requirements:

"We thank Shigeki Higashiyama (Ehime University) for the AP-fused constructs; Makoto Arita (Keio University) for the helpful information on the TRP channel and TGF-α shedding response; Kayo Sato, Shigeko Nakano and Ayumi Inoue (Tohoku University) for their assistance in plasmid preparation. A.I. was supported by Japan Society for the Promotion of Science (JSPS) KAKENHI grants 21H04791, 21H05113, JPJSBP120213501 and JPJSBP120218801; FOREST Program JPMJFR215T and JST Moonshot Research and Development Program JPMJMS2023 from Japan Science and Technology Agency (JST); Takeda Science Foundation; Uehara Memorial Foundation. M.T. received JSPS KAKENHI 22J10475. The funders had no role in study design, data collection and analysis, decision to publish, or preparation of the manuscript."

"A.I. was supported by Japan Society for the Promotion of Science (JSPS) KAKENHI grants H04791, 21H05113, JPJSBP120213501 and JPJSBP120218801; FOREST Program JPMJFR215T and JST Moonshot Research and Development Program JPMJMS2023 from Japan Science and Technology Agency (JST); Takeda Science Foundation; Uehara Memorial Foundation. M.T. received JSPS KAKENHI 22J10475. The funders had no role in study design, data collection and analysis, decision to publish, or preparation of the manuscript."

Reviewers' comments:

Reviewer's Responses to Questions

**Comments to the Author**

1. Is the manuscript technically sound, and do the data support the conclusions?

Reviewer #1: Yes

Reviewer #2: Yes

2. Has the statistical analysis been performed appropriately and rigorously? 

Reviewer #1: Yes

Reviewer #2: Yes

3. Have the authors made all data underlying the findings in their manuscript fully available?

Reviewer #1: Yes

Reviewer #2: Yes

4. Is the manuscript presented in an intelligible fashion and written in standard English?

Reviewer #1: Yes

Reviewer #2: Yes

5. Review Comments to the Author

Reviewer #1: In this work, Tatsumi et al. developed a new molecular biology method to assay TRP channels activity exogenously expressed in cell lines, based on the use of a transforming TGF alpha shedding assay. This assay has the advantage of being low cost compared to calcium imaging or patch-clamp tecniques. That said, the information obtained by this method doesnt´t allow to monitorize TRP channel activity since the measurement is based on cummulated activity. Further, it is not useful to evaluate the activity of natively expressed TRP channels. Despite those limitations, this work might help the high-throughput screening of TRP channels ligand and to clarify the relationship between TRP channels and patho-physiologically relevant proteins as ADAM. Overall, the manuscript is well-written, well-structured, and the data support the conclusions.

As a main concern, authors show spontaneous TRPV1 activation (Figure 1H), however the origin of this activity remains unclear in the manuscript, since authors do not discuss this point or mention any reference supporting this. Is this TRPV1 spontaneous activation developed as a consequence of TGF alpha transfection? Has been this spontaneous activity previously reported in calcium imaging or patch-clamp experiments? If so, authors should add the corresponding references to the draft.

Reviewer #2: The present manuscript by Tatsumi and colleagues is an attempt to develop a low-cost and easy-to-access method for evaluating TRP channel activation. This will be useful for researchers, especially those who have limited access to specialized equipment, and may broaden the TRP channel field.

The present manuscript should be accepted in its present form.

6. PLOS authors have the option to publish the peer review history of their article (what does this mean?). If published, this will include your full peer review and any attached files.

Reviewer #1: No

Reviewer #2: No

---

## [Author Response · Author response to Decision Letter 0]

21 Dec 2022

Our point-to-point responses are included in the separately uploaded response letter.

---

## [Editor Report · Decision Letter 1]

2 Jan 2023

Ectodomain shedding of EGFR ligands serves as an activation readout for TRP channels

PONE-D-22-29508R1

Dear Dr. Inoue,

We’re pleased to inform you that your manuscript has been judged scientifically suitable for publication and will be formally accepted for publication once it meets all outstanding technical requirements.

Kind regards,

Agustín Guerrero-Hernandez

Academic Editor

PLOS ONE
---

## [Editor Report · Acceptance letter]

11 Jan 2023

PONE-D-22-29508R1 

Ectodomain shedding of EGFR ligands serves as an activation readout for TRP channels 

Dear Dr. Inoue:

I'm pleased to inform you that your manuscript has been deemed suitable for publication in PLOS ONE. Congratulations! Your manuscript is now with our production department. 

Kind regards, 

on behalf of

Dr. Agustín Guerrero-Hernandez 

Academic Editor

PLOS ONE